# Robotic-Assisted Thoracoscopic (RATS) Enucleation of Esophageal Mesenchymal Tumors and Foregut Cysts

**DOI:** 10.3390/jcm11216471

**Published:** 2022-10-31

**Authors:** Yung-Hsin Chen, Ke-Cheng Chen, Pei-Ming Huang, Shuenn-Wen Kuo, Mong-Wei Lin, Jang-Ming Lee

**Affiliations:** Department of Surgery, National Taiwan University Hospital, Taipei 100225, Taiwan

**Keywords:** esophageal tumor, leiomyoma, gastrointestinal stromal tumor, robotic surgery, enucleation

## Abstract

**Background:** Esophageal mesenchymal tumors and foregut cysts are mostly benign lesions of the esophagus. Tumor enucleation is recommended for lesions with a risk of malignancy, or for the relief of clinical symptoms. Although robotic-assisted thoracoscopic enucleation of esophageal tumors and cysts has been demonstrated in sporadic case reports, its clinical role is yet to be elucidated. **Methods**: This study aimed to present the first case series in the literature for the perioperative and long-term clinical outcomes of robotic-assisted thoracoscopic enucleation. **Results**: A total of 19 patients who underwent robotic-assisted thoracoscopic enucleation of esophageal tumors and cysts from 2012 to 2019 were included in the study. The mean tumor/cyst size was 5.5 cm (1.5–22 cm). There were two cases shifting to minimally invasive esophagectomy (10.5%) due to intraoperative pathological confirmation of malignant gastrointestinal stromal tumors with mucosal invasion. Perioperative complication was detected in three (15.8%) cases, without 30-day or surgical mortality. There was no recurrence of tumor or symptoms in all patients during the clinical follow-up period (mean = 35 months). **Conclusions**: Robotic-assisted thoracoscopic enucleation of esophageal submucosal benign tumors is technically feasible and effective. Given its advantage in overcoming spatial limitations, it can become a widely accepted surgical option for such diseases.

## 1. Introduction

Esophageal mesenchymal tumors are rare lesions, which account for less than 1% of all esophageal neoplasms, with leiomyoma and gastrointestinal stromal tumors (GISTs) occurring in most cases [1]. Surgical resection is generally indicated for diagnosing and treating symptomatic tumors or tumors larger than 5 cm [2,3]. Some studies also suggest surgery for tumors larger than 1 cm [4].

Esophageal foregut cysts or duplication cysts are rare congenital anomalies. They are mostly asymptomatic in adult patients, and surgical enucleation is suggested to prevent cyst rupture, infection, bleeding, and rare malignant transformation [5,6].

Robotic-assisted thoracoscopic enucleation was first reported by Elli et al. in 2004 [3]. The authors emphasized overcoming the spatial limitation posed by video-assisted thoracoscopic surgery while preserving the benefits of minimal invasiveness. Although it has been attempted by several reports for esophageal mesenchymal tumors of different sizes and locations (Table 1) [7,8,9], its clinical value has not been fully understood in the literature. This study aimed to present the first case series in the literature for the perioperative and long-term clinical outcome of robotic-assisted thoracoscopic enucleation.

## 2. Patients and Methods

Patients with esophageal mesenchymal tumors or cysts undergoing robotic-assisted thoracoscopic enucleation from 2012 to 2019 were evaluated retrospectively. They were all operated on by a single surgeon in our institute. The preoperative imaging studies included upper gastrointestinal endoscopy, computed tomography (CT), and endoscopic ultrasound (EUS). Biopsy was not performed if benign esophageal tumors were suggested. Surgery was recommended to the patients if the tumor/cyst size was larger than 1 cm or the lesion was symptomatic [4]. Tumor enucleation was attempted for all of the lesions without any clinical evidence of mucosal invasion by the tumors. Esophagectomy with esophageal reconstruction would be performed if mucosal involvement by the tumor was detected during surgery. Demographic data, tumor size, tumor location, symptoms, pathology, operative approach, operation time, and short-term and long-term complications were all recorded and analyzed. This study has been approved by the Research Ethics Committee review board of the hospital (202201100RIND).

After general anesthesia with double-lumen intubation, the right or left semi-prone position was adopted according to the laterality of the tumor in the esophagus in the imaging study. The three-arm da Vinci Si or Xi surgical system (Intuitive Surgical Inc., Sunnyvale, CA, USA) was used.

The camera port was inserted in the 6th–8th intercostal space (ICS) at the midaxillary line. Two main ports were set in the 6th–8th ICS at the anterior axillary line, and the 7th–9th ICS at the posterior axillary line, with the bipolar forceps (Fenestrated Bipolar Forceps, Intuitive Surgical Inc., Sunnyvale, CA, USA) and the unipolar dissector (Permanent Cautery Spatula, Intuitive Surgical Inc., Sunnyvale, CA, USA) placed, respectively. Optional assistant ports were created in the 9th–10th ICS before the anterior axillary line.

There were three major steps in our surgery (Figure 1). First, we retracted the lung to expose the esophagus, and circumferentially mobilized the esophagus after incising the mediastinal pleural and dividing the esophageal adventitia. The azygos vein was divided by an endo cutter if the tumor/cyst was at the middle or upper esophagus. The inferior pulmonary ligament was divided if the tumor/cyst was at the lower esophagus. Second, the tumor/cyst was visualized after retracting the mobilized esophagus. We performed longitudinal myotomy to expose the tumor/cyst. Blunt dissection was utilized to divide the tumor from the submucosal tissue. Enucleation was completed with the mucosa preserved. Last, the muscular layer was reapproximated with a V-loc or polydioxanone (PDS) running suture to prevent diverticulum formation. The wounds were closed in layers, with a chest tube placed through the camera port. A frozen section was performed if the tumor invaded the mucosa. Subtotal esophagectomy was performed for pathologically-confirmed GISTs. Regional lymph nodes in the thoracic paraesophageal areas were also dissected for staging for these patients [25,26].

## 3. Results

There were 19 patients (11 males and 8 females) in this case series (Table 2). The median age was 56 years (range 34–86). Only eight patients were symptomatic, with dysphagia (50.0%) being the most common symptom. Chest discomfort was also noted in two patients (25.0%). Other asymptomatic patients were diagnosed by either health examination or incidental findings from CT or endoscopy.

The mean tumor/cyst size was 5.5 cm (range 1.5–22). In total, five (26.3%) were larger than 5 cm. Most of the tumors and cysts were located in the lower esophagus (57.9%). A total of four (36.4%) patients presented with upper esophagus tumors. Another four (36.4%) presented with middle esophagus tumors. Leiomyoma was confirmed in 10 (52.6%) patients, and GIST was found in four (36.4%) patients. Schwannoma (10.5%), lipoma (5.3%), granular cell tumor (5.3%), and foregut cyst (5.3%) were also reported. 

The enucleation of esophageal tumors and cysts were accomplished in all cases. However, two cases (10.5%) were found to have mucosal invasion during the operation. Frozen pathology confirmed the diagnosis of GISTs. Therefore, the operation was converted to robotic-assisted thoracoscopic subtotal esophagectomy and reconstruction with laparoscopic gastric tube formation (Ivor Lewis esophagectomy). The median operation time was 99 min (range 71–247). The median docking time was 9 min (range 5–55), whereas the median console time was 83 min (range 38–242). Blood loss was mostly minimal. An increased amount of blood loss was noted in esophagectomy cases, with transfusion required. Two representative cases are demonstrated in Figure 1 and Figure 2.

The median length of hospital stay was 11 days (range 5–53). Post-operative complications were noted in three (15.8%) patients: one with sepsis, another with chylothorax, and the third with GI bleeding, hiatal hernia, and stroke. There was no mortality. The median interval between surgery and liquid intake was 5 days after water-soluble contrast medium (hypaque) swallow showed no leakage. A soft diet was then tried the next day, and chest tubes were withdrawn on the third day.

Imatinib mesylate (Glivec) was administered to patients with pathologically confirmed GISTs for disease control. The median follow-up time was 35 months (range 1–74). No long-term complications were observed, including tumor recurrence, diverticulum, or stricture.

## 4. Discussion

Our study is the largest case series to date on robotic-assisted thoracoscopic enucleation of esophageal mesenchymal tumors and cysts. There have been 35 cases reported to date (Table 1) [3,6,7,8,10,11,12,13,14,15,16,17,18,19,20,21]. In total, 22 (62.8%) patients were diagnosed with leiomyoma, 6 (17.1%) patients had foregut cysts, and 1 (2.8%) patient had schwannoma. The tumor size ranged from 2 cm to 10 cm, with the locations distributed evenly from the upper to lower esophagus. Only one patient had complications: an esophageal fistula after the operation. No intraoperative mucosal damage was described.

Similar to the previous data, we presented 19 cases, with more than half being leiomyoma. However, apart from foregut cyst and schwannoma, we also reported four cases of malignant GISTs, one lipoma, and one granular cell tumor. We had a wider tumor size range (1.5 cm–22 cm). More than half of them were located in the lower esophagus. Intraoperative mucosal damage was noted in two patients with GISTs adhesive to the mucosa. Enucleations were all converted to subtotal esophagectomy for malignancy treatment.

Leiomyoma is the most common esophageal mesenchymal tumor, which accounts for about 70–80% of all esophageal mesenchymal tumors. It is hard to distinguish it from GISTs through preoperative imaging, accounting for another 15–25% of esophageal mesenchymal tumors [27]. Other rare entities, including schwannomas, hemangioma, lipomas, leiomyosarcoma, papillary epithelioma, and granular cell tumors, account for less than 5% of all esophageal mesenchymal tumors [28].

It is essential to differentiate GISTs from leiomyoma, since GISTs are considered malignant. However, this differentiation cannot be done using CT, EUS, and 18F-fluorodeoxyglucose positron emission tomography [29]. The golden standard for differentiation is a pathological examination, including hematoxylin and eosin staining, and immunohistochemical studies [30]. To obtain enough tissue for diagnosis, core biopsy and fine needle aspiration were performed in some cases. However, they are not recommended due to the increased risk of operative mucosal perforation caused by fibrosis and the adhesion between the mucosa and the tumors [31]. Moreover, fine needle aspiration is insufficient to provide enough tissue for differentiation, and risks tumor seeding [32].

Therefore, surgical enucleation for symptomatic esophageal mesenchymal tumors or those larger than 5 cm is recommended by most studies for definite diagnosis and treatment [2,3]. For esophageal mesenchymal tumors smaller than 1 cm, observation is recommended. Most studies preferred an elective minimal invasive surgery for tumors in between to prevent the increased morbidity and invasiveness of large tumor excision [4].

Enucleation for esophageal mesenchymal tumors is traditionally performed by thoracotomy. After the first reported video-assisted thoracoscopic enucleation in 1992 by Everitt [33], it is widely accepted due to less invasiveness, reduced length of hospital stay, decreased pain score, less morbidity, and less mortality. However, for upper esophageal tumors with larger sizes (>5 cm), spatial limitation increased the risk of morbidity, mucosal damage, and conversion [4].

Robotic-assisted thoracoscopic enucleation was then promoted for its three-dimensional magnified vision, ergonomic comfort, motion scaling, and tremor filtration, which might help reduce the risk of surgical complications and intraoperative esophageal mucosal injury [16]. If we take our study into consideration to make a total of 42 patients who underwent robotic surgery, the mucosal injury rate is 4.8%, whereas the complication rate is 9.5%. Whether it can provide additional benefit compared to VATS or open surgery remains to be elucidated in the future.

The main limitations of our study were the small patient number and variations of the tumor characteristics. Nevertheless, this is the largest series in the literature of robotic-assisted tumor excision for esophageal submucosal lesions. The perioperative outcome and long-term follow-up result seemed satisfactory in our patients. In the future, multi-center studies could be conducted to evaluate the optimal indication and clinical value of robotic-assisted surgery for esophageal submucosal tumors.

## 5. Conclusions

Robotic-assisted thoracoscopic enucleation of esophageal mesenchymal tumors and foregut cysts is technically safe and effective. Given its advantage in overcoming spatial limitations, it has the potential to become a preferred surgical option for esophageal mesenchymal tumors and foregut cysts. Further larger-scale multi-center studies are needed to provide a more concrete conclusion.

## Figures and Tables

**Figure 1 jcm-11-06471-f001:**
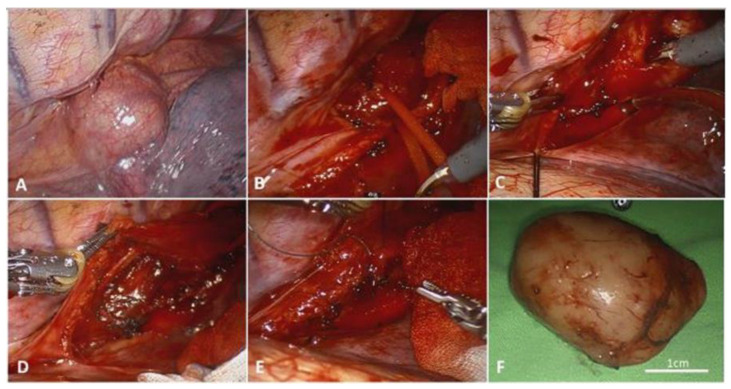
Case 15. (**A**) Esophageal mesenchymal tumor at the lower thorax. (**B**) Retracting the mobilized esophagus. (**C**) Longitudinal myotomy and tumor excision at the submucosal layer. (**D**) No mucosal perforation was noted. (**E**) Repairing the myotomy with 3-0 V-loc. (**F**) A 4 cm well circumscribed tumor, pathology confirmed GIST, and Glivec was administered.

**Figure 2 jcm-11-06471-f002:**
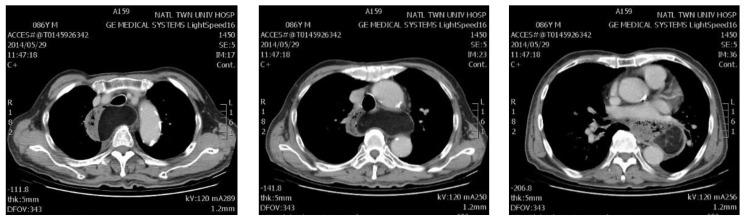
The CT (Computed Tomography) scan of the patient with giant lipoma of 22 cm in diameter (case 4). He received robotic-assisted tumor excision and was discharged 12 days after surgery.

**Table 1 jcm-11-06471-t001:** Literature review of robotic-assisted thoracoscopic enucleation.

Author	N	Location	Size (cm)	Pathology	Operation Time (Min)	Complications	Hospital Stay (d)
Elli et al. [3]	2	Ut ^◎^, Mt ^☆^	5, 3	Leiomyoma (2)	120	None	NA
Bodner et al. [10]	2	Lt ^★^	6, 5	Leiomyoma, foregut cyst	147, 95	None	7
Augustin et al. [11]	2	NA ^△^	2, 5.5	Leiomyoma, foregut cyst	147, 151	None	4
Boone et al. [12]	1	Ut ^◎^	9	Leiomyoma	270	None	11
DeUgarte et al. [13]	1	Mt ^☆^	7	Leiomyoma	NA	None	5
Kernstine et al. [14]	1	Lt ^★^	3	Leiomyoma	104	None	1
Ka-fung chiu et al. [15]	1	Mt ^☆^	2	Leiomyoma	NA	None	6
Obasi et al. [6]	2	Mt ^☆^	2	Foregut cyst	NA	None	2, 3
Khalaileh et al. [16]	1	Lt ^★^	5	Leiomyoma	288	None	3
Compean et al. [7]	1	Ut ^◎^	10	Leiomyoma	NA	None	3
Zhang et al. [17]	1	Lt ^★^	7	Schwannoma	108	None	5
Tomulescu et al. [18]	4	Lt ^★^	3–5	Leiomyoma (3), foregut cyst (1)	195 ^#^ (150–240)	Fistula (1)	6 ^#^ (5–21)
Ramírez et al. [19]	1	Ut ^◎^	4.1	Leiomyoma	NA	None	1
Inderhees et al. [20]	1	Ut ^◎^	5.5	Leiomyoma	143	None	5
Elliott et al. [21]	1	Mt^☆^	2.4	Leiomyoma	NA	None	2
Kemuriyama et al. [8]	1	Ut ^◎^-Mt ^☆^	10	Leiomyoma	329	None	4
Froiio et al. [22]	6	Ut ^◎^-Lt ^★^	2.8–7.7	GIST(3) ^◆^ Leionyoma(3),	154 (129–232)	Pneumonia(1), delayed gastric emptying(1)	7 (6–500)
Yamamoto et al. [23]	1	Mt ^☆^	3.0	GIST ^◆^	319	None	18 (RAMIE ^▲^)
Tribuzi et al. [24]	5	Mt ^☆^-Lt ^★^	3.7 (30–63)	GIST(2) ^◆^, Leionyoma(2), GIST(1) ^◆^	150 (100–300)	None	5 (4–9)
Our study	19	Ut (4) ^◎^, Mt (4) ^☆^, Lt (11) ^★^	5.5 * (1.5–22)	Leiomyoma (10), GISTs ^◇^ (4) ^◆^, schwannoma (2), lipoma (1) [9], granular cell tumor (1), foregut cyst (1)	99 ^#^ (71–247)	Sepsis (1), chylothorax (1), GI ^□^ bleeding/hiatal hernia/stroke (1)	11 ^#^ (5–53)

* mean tumor size, ^#^ median operation time or hospital stay, ^△^ NA: not applicable, ^▲^ RAMIE: robotic assisted minimally invasive esophagectomy, ^□^ GI: gastro-intestinal, ^◎^ Ut: upper-third tumor, ^☆^ Mt: middle-third tumor, ^★^ Lt: lower-third tumor, ^◆^ GIST: gastrointestinal stromal tumor, ^◇^ GISTs: plural of GIST.

**Table 2 jcm-11-06471-t002:** Case series presentation.

Case	Age (Years)/Sex	Location	Size (cm)	Symptoms	Pathology	Operation Time (Min)	Hospital Stay (d)
1	64/M	Mt ^☆^	1.5	None	Leiomyoma	82	8
2	53/F	Ut ^◎^	3	None	Leiomyoma	103	10
3	34/M	Lt ^★^	5	None	Leiomyoma	NA	11
4	86/M	Lt ^★^	22	Dysphagia	Lipoma	151	12
5	60/F	Ut ^◎^	4	None	Leiomyoma	136	8
6	36/M	Ut ^◎^	6	Chest pain	Leiomyoma	85	13
7	34/M	Lt ^★^	12	Dysphagia	GIST ^◆^	247 *	19
8	47/F	Mt ^☆^	3	None	Leiomyoma	76	16
9	63/F	Lt ^★^	4	None	Leiomyoma	91	10
10	40/F	Lt ^★^	5	None	Leiomyoma	94	28
11	77/F	Lt ^★^	10	None	Schwannoma	197	13
12	38/M	Lt ^★^	4	Epigastric pain	GIST ^◆^	NA	34
13	42/M	Lt ^★^	2.5	Hemoptysis	Granular cell tumor	71	11
14	60/M	Mt ^☆^	2	None	Leiomyoma	NA	8
15	63/M	Lt ^★^	4	None	GIST ^◆^	119	10
16	67/M	Lt ^★^	10	Dysphagia	GIST ^◆^	241 *	53
17	56/F	Mt ^☆^	2.5	Dysphagia	Schwannoma	81	10
18	55/F	Ut ^◎^	2	Hoarseness	Leiomyoma	NA	5
19	60/M	Lt ^★^	2.5	None	Foregut cyst	NA	8

* subtotal esophagectomy performed, ◎ Ut: upper-third tumor, ◆ GIST: gastrointestinal stromal tumor, ☆ Mt: middle-third tumor, ★ Lt: lower-third tumor.

## Data Availability

Not applicable.

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
