# Peer review of "Robotic-Assisted Thoracoscopic (RATS) Enucleation of Esophageal Mesenchymal Tumors and Foregut Cysts"

_jcm, 2022, doi:10.3390/jcm11216471_

Round 1

Reviewer 1 Report

This is a single-center retrospective analysis of a case-series following robot-assisted resection of esophageal mesenchymal tumors. The authors present their results after surgical resection of a variety of mesenchymal tumors and cysts of the esophagus applying the robotic approach. However, the present manuscript’s significant issues require major revision.

       I.          This is not a novel study, as several previous publications have already presented their outcomes. It goes without saying that these should be mentioned and underlined in this study and, of course, this stands among the major disadvantages of the paper (lack of novelty, does not add much information on the existing literature). Looking at the recent references, not one of the existing case-series has been reported by the authors. i.e.

1.      Froiio C, Berlth F, Capovilla G, Tagkalos E, Hadzijusufovic E, Mann C, Lang H, Grimminger PP. Robotic-assisted surgery for esophageal submucosal tumors: a single-center case series. Updates Surg. 2022 Jun;74(3):1043-1054. doi: 10.1007/s13304-022-01247-z. Epub 2022 Feb 11. PMID: 35147859; PMCID: PMC9213313.

2.      Yamamoto H, Ebihara Y, Tanaka K, Matsui A, Nakanishi Y, Asano T, Noji T, Kurashima Y, Murakami S, Nakamura T, Tsuchikawa T, Okamura K, Shichinohe T, Hirano S. Robot-assisted thoracoscopic esophagectomy for gastrointestinal stromal tumor of the esophagus: A case report. Int J Surg Case Rep. 2021 Sep;86:106335. doi: 10.1016/j.ijscr.2021.106335. Epub 2021 Aug 26. PMID: 34481133; PMCID: PMC8416945.

3.      Tribuzi A, Bencini L, Paolini C, Di Marino M, Coratti A. Robotic enucleation for oesophageal benign and borderline tumours: Less is more? Int J Med Robot. 2021 Feb;17(1):1-7. doi: 10.1002/rcs.2178. Epub 2020 Oct 12. PMID: 33010797.

     II.          Furthermore, in the introduction, the indications for resection of these tumors are confusing. The authors state “Surgical resection is generally indicated for diagnosing and treating symptomatic tumors or tumors larger than 5 cm” but in their methods they have included asymptomatic patients with smaller lesions.

Please clarify based on current guidelines.

   III.          EUS applied in the preoperative setting is of paramount importance. Although the cases included underwent preoperative EUS, two of the cases eventually underwent esophagectomy as an intraoperative decision making. Was preop.EUS not diagnostic for mucosal invasion? That is a significant change of the surgical plan and should be predetermined through accurate preop.imaging and not based on frozen section.

  IV.          The manuscript needs major revision for typos and grammar inaccuracies. i.e.

1.      Abstract: “Although robotic-assisted thoracoscopic enucleation of esophageal tumors and cysts have been demonstrated in sporadic case reports, its clinical role is yet…”

2.       Introduction: “They are 33 mostly asymptomatic in adult patients, and surgical enucleation is suggested to prevent 34 cyst rupture, infection, bleeding, and rare malignant transformation”

3.      Introduction: “Although it has 38 been attempted severally…”

4.      Methods “Biopsy was not performed if benign esophageal tumors were suggested…”

    V.          It may be worth presenting a picture from the case diagnosed with the largest lesion (22cm lipoma)

  VI.          The term “dividing the esophageal serosa” is inaccurate, please clarify the esophageal layers

VII.          Lymphadenectomy is not applied for the resection of these tumors “Regional lymph nodes were also dissected for staging”. Why did the surgeon decide to proceed with lymph node dissection?

VIII.          The postoperative complications are not clearly reported. CLavien-Dindo classification would be very useful. Moreover, the hospital stay is very prolonged in many of the cases. Please comment on the reasons as this is a major issue taking into consideration the morbidity reported following standard resection of mesenchymal tumors and benign esophageal lesions.

  IX.          Immunohistology of the tumors resected could be reported. The role of adjuvant imatinib is neither reported in the text.

    X.          The tables’ abbreviations are not clarified.

Author Response

Reviewer 1

  1. This is not a novel study, as several previous publications have already presented their outcomes. It goes without saying that these should be mentioned and underlined in this study and, of course, this stands among the major disadvantages of the paper (lack of novelty, does not add much information on the existing literature). Looking at the recent references, not one of the existing case-series has been reported by the authors. i.e.

  1. Froiio C, Berlth F, Capovilla G, Tagkalos E, Hadzijusufovic E, Mann C, Lang H, Grimminger PP. Robotic-assisted surgery for esophageal submucosal tumors: a single-center case series. Updates Surg. 2022 Jun;74(3):1043-1054. doi: 10.1007/s13304-022-01247-z. Epub 2022 Feb 11. PMID: 35147859; PMCID: PMC9213313.

  1. Yamamoto H, Ebihara Y, Tanaka K, Matsui A, Nakanishi Y, Asano T, Noji T, Kurashima Y, Murakami S, Nakamura T, Tsuchikawa T, Okamura K, Shichinohe T, Hirano S. Robot-assisted thoracoscopic esophagectomy for gastrointestinal stromal tumor of the esophagus: A case report. Int J Surg Case Rep. 2021 Sep;86:106335. doi: 10.1016/j.ijscr.2021.106335. Epub 2021 Aug 26. PMID: 34481133; PMCID: PMC8416945.

3.Tribuzi A, Bencini L, Paolini C, Di Marino M, Coratti A. Robotic enucleation for oesophageal benign and borderline tumours: Less is more? Int J Med Robot. 2021 Feb;17(1):1-7. doi: 10.1002/rcs.2178. Epub 2020 Oct 12. PMID: 33010797.

Response:

Thank you so much for the suggestion.  We have added these references into the manuscript in Table 1 and described further in lines 133-135.

  1. Furthermore, in the introduction, the indications for resection of these tumors are confusing. The authors state “Surgical resection is generally indicated for diagnosing and treating symptomatic tumors or tumors larger than 5 cm” but in their methods they have included asymptomatic patients with smaller lesions.

Please clarify based on current guidelines.

Response:

Although many authors agree with that tumor with size more than 5 cm requires definitive surgical resection. With the approach of minimally invasive surgery, tumor can be removed earlier than that status with less surgical trauma while without the risk of complications encountered during handling huge tumor.  This has been discussed in lines 161-165.    

III. EUS applied in the preoperative setting is of paramount importance. Although the cases included underwent preoperative EUS, two of the cases eventually underwent esophagectomy as an intraoperative decision making. Was preop.EUS not diagnostic for mucosal invasion? That is a significant change of the surgical plan and should be predetermined through accurate preop.imaging and not based on frozen section.

Response:

We have used EUS for every patient once esophageal submucosal tumor was impressed.  The tissue diagnosis detection and of mucosal invasion for the submucosal tumor of the esophagus might be difficult in some patients. For such uncertain cases, we would try to preserve the esophagus and enucleation of the tumor at first.  Esophagectomy will be performed if such attempt is difficult during the surgery.  We have added further description about the decision making for the patients about the surgical plan in the manuscript between lines 51-54. 

  1. The manuscript needs major revision for typos and grammar inaccuracies. i.e.

  1. Abstract: “Although robotic-assisted thoracoscopic enucleation of esophageal tumors and cysts have been demonstrated in sporadic case reports, its clinical role is yet…”

2.Introduction: “They are 33 mostly asymptomatic in adult patients, and surgical enucleation is suggested to prevent 34 cyst rupture, infection, bleeding, and rare malignant transformation”

3.Introduction: “Although it has 38 been attempted severally…”

4.Methods “Biopsy was not performed if benign esophageal tumors were suggested…”

Response:

The typos and grammar inaccuracies have been corrected as suggested.  

V.It may be worth presenting a picture from the case diagnosed with the largest lesion (22cm lipoma)

Response:

The picture has been shown in the Figure 2 and described in line 117.  

VI.The term “dividing the esophageal serosa” is inaccurate, please clarify the esophageal layers

Response:

Thanks for the correction.  We have changed the terminology as “ dividing the esophageal adventitia” in line 70. 

VII. Lymphadenectomy is not applied for the resection of these tumors “Regional lymph nodes were also dissected for staging”. Why did the surgeon decide to proceed with lymph node dissection?

Response:

Lymphadenectomy for accurate staging of tumor was performed only for the patients when a diagnosis of malignancy during surgery (GIST) was suspected.  It has been described in lines 80-81.

VIII. The postoperative complications are not clearly reported. CLavien-Dindo classification would be very useful. Moreover, the hospital stay is very prolonged in many of the cases. Please comment on the reasons as this is a major issue taking into consideration the morbidity reported following standard resection of mesenchymal tumors and benign esophageal lesions.

Response:

We have performed esophagogram before resumption of the oral intake for the patients.   Due to the schedule arrangement, the hospital stay might be therefore prolonged after completion of the examination. However, there were four patients with admission for more than or around three weeks after surgery due to esophagectomy (1), catheter-related spepsis (1), chylothorax (1) and hiatal hernia (1). All of the complications have been described in lines 122-123.    

  1. Immunohistology of the tumors resected could be reported. The role of adjuvant imatinib is neither reported in the text.

Response:

For the patients with GIST revealed by the pathological report, prolonged use of imatinib would be given in the out-patient clinics of oncologist.  It has been described lines 127-8. 

  1. The tables’ abbreviations are not clarified.

Response:

More description has been given in the tables.  

Reviewer 2 Report

This study describes a substantial series for this condition and treatment approach of patients from a single institution and a single surgeon. It almost doubles the numbers of reported cases previously and therefore holds merit. Most important message is that a minimally invasive approach to these lesions is promoted which is justified by the results. 

A few adjustments are needed: The last statement of the methods section should be specified better or removed. Lymph node dissection in general for these lesions is not evidence based. The two papers cited to support this (10 and 11) actually do not support this! When esophagectomy is performed in case of the largers GISTs, regional lymph node stations will automically come with the specimen of course. But when unucleating a small lesion less that 10cm, I think removing lymph nodes is contra-indicated and will only add morbidity. What stations should be taken? To what extend??

Why weren´t there any more recent cases added and does the series stop in 2019?

Please avoid data description in the discussion (first paragraph), should be transferred to the results section.

Author Response

 Reviewer 2

I A few adjustments are needed: The last statement of the methods section should be specified better or removed. Lymph node dissection in general for these lesions is not evidence based. The two papers cited to support this (10 and 11) actually do not support this! When esophagectomy is performed in case of the largers GISTs, regional lymph node stations will automically come with the specimen of course. But when unucleating a small lesion less that 10cm, I think removing lymph nodes is contra-indicated and will only add morbidity. What stations should be taken? To what extend??

Response

Lymphadenectomy around the thoracic paraesophageal regions was performed only for the patients when a diagnosis of malignancy during surgery (GIST) was suspected.  It has been described in lines 80-81.

  1. Why weren´t there any more recent cases added and does the series stop in 2019?

Response:

The surgery is conducted continuously in our institute.  However, the study cohort was collected till 2019 for observation of the mid term outcome. 

Please avoid data description in the discussion (first paragraph), should be transferred to the results section.

Response:

Thank you for the suggestion. The description has been made for the results of literature review which is therefore remained in the discussion section instead of results section.  

Round 2

Reviewer 1 Report

I am happy to accept the revised manuscript for publication in its current form.